# Fabrication of Compliant and Transparent Hollow Cerebral Vascular Phantoms for In Vitro Studies Using 3D Printing and Spin–Dip Coating

**DOI:** 10.3390/ma16010166

**Published:** 2022-12-24

**Authors:** Beatrice Bisighini, Pierluigi Di Giovanni, Alba Scerrati, Federica Trovalusci, Silvia Vesco

**Affiliations:** 1Mines Saint-Etienne, Université Lyon, Université Jean Monnet, Etablissement Français du Sang, INSERM, U1059 Sainbiose, Centre CIS, F-42023 Saint-Etienne, France; beatrice.bisighini@emse.fr; 2Department of Enterprise Engineering, University Tor Vergata, Via del Politecnico 1, 00133 Rome, Italy; federica.trovalusci@uniroma.eu (F.T.); silvia.vesco@uniroma2.it (S.V.); 3Predisurge, 10 Rue Marius Patinaud, Grande Usine Creative 2, 42000 Saint-Etienne, France; 4HSL s.r.l., Via dei Masadori 46, 38121 Trento, Italy; pierluigi.digiovanni@consultants.hsl-italia.com; 5Department of Translational Medicine, University of Ferrara, Via Luigi Borsari 46, 44121 Ferrara, Italy

**Keywords:** 3D printing, cerebral aneurysms, endovascular, coating

## Abstract

Endovascular surgery through flow diverters and coils is increasingly used for the minimally invasive treatment of intracranial aneurysms. To study the effectiveness of these devices, in vitro tests are performed in which synthetic vascular phantoms are typically used to reproduce in vivo conditions. In this paper, we propose a manufacturing process to obtain compliant and transparent hollow vessel replicas to assess the mechanical behaviour of endovascular devices and perform flow measurements. The vessel models were obtained in three main steps. First, a mould was 3D-printed in a water-soluble material; two techniques, fusion deposition modelling and stereolithography, were compared for this purpose. Then, the mould was covered with a thin layer of silicone through spin–dip coating, and finally, when the silicone layer solidified, it was dissolved in a hot water bath. The final models were tested in terms of the quality of the final results, the mechanical properties of the silicone, thickness uniformity, and transparency properties. The proposed approach makes it possible to produce models of different sizes and complexity whose transparency and mechanical properties are suitable for in vitro experiments. Its applicability is demonstrated through idealised and patient-specific cases.

## 1. Introduction

Nowadays, realistic vessel replicas are emerging as a new paradigm for the design and testing of medical devices used for the endovascular treatment of cerebral aneurysms (CAs), such as coils and flow diverters [1]. The effectiveness of these devices is measured in terms of their ability to isolate the aneurysmal sac from the bloodstream without compromising the parent artery. Therefore, in vitro analyses must take into account both the flow changes induced by the presence of the device and the mechanical forces exerted on the vessel wall; gaining a better understanding of these mechanisms could improve CA endovascular treatment.

Once deployed within 3D anatomical phantoms, the geometry of these devices can be obtained from 3D images through segmentation. However, their small size and the artefacts induced by the metallic structure make surface reconstruction very difficult, if not impossible, with conventional medical imaging equipment. Thus, high-spatial-resolution imaging techniques, such as micro-computed tomography (micro-CT), should be considered [2,3,4,5]. On the other hand, blood hemodynamics both before and after the endovascular intervention can be accessed using phase contrast magnetic resonance imaging (PC-MRI) [6,7,8] or optical flow measurement techniques, e.g., particle image velocimetry (PIV) and laser Doppler velocimetry (LDV) [9,10,11]. These studies are considered the most physiologically representative ground truth for the validation of in silico simulations [12,13,14,15].

To be used for such optical studies, the models should not introduce any distortion in the recorded signal. This is always the case when a light signal passes through a curved wall. Since no straightforward post-processing correction is possible for complex vessel geometries, the best solution to avoid reconstruction errors is to choose a working fluid and a model material that have nearly identical refractive indexes (refractive index matching principle) [16].

Advances in 3D printing have made it possible to build models that mimic the structure and mechanical behaviour of patient-specific arteries for in vitro studies. To date, the techniques for building compliant vessel models can be placed in two main categories: direct and indirect additive manufacturing (AM). The former is most commonly based on Polyjet 3D printing [17], for which several elastic photopolymerisable resins are available in the market, e.g., Agylus30 [18] and TangoPlus FLX930 [19]. Direct AM allows obtaining controlled-thickness models with mechanical properties very close to the biological ones in an easier and faster way than conventional indirect methods [20,21]. However, matching the refractive index (RI) of the final models with the most common working fluids can only be achieved for rigid phantoms with direct AM [22,23,24]; in fact, soft resins currently available on the market have a high RI, as reported in [16].

Compared with resins, silicones present a lower RI and, therefore, have been widely used to build vessel phantoms by indirect printing [25]. Several indirect 3D printing techniques have been proposed in the literature. Casting is the most prevalent method and is based on the use of an inner and outer mould; the liquid material is poured into the space between these two moulds and solidifies into a controlled-thickness model [26,27] Another well-established option is dip or dip–spin coating, which does not require the use of an outer mould; the inner mould is dipped in a silicone bath to create a thin silicone film on the surface and, eventually, connected to a rotation system to evenly distribute the silicone [28]. As an alternative, the inner mould can be painted to reduce the amount of residual silicone [29,30]. To facilitate the extraction of the model once cured, moulds are made of mechanically removable or dissolvable materials. The former, e.g., wax, requires heating and/or mechanical force that can damage the structure [31,32,33]. On the other hand, dissolvable moulds involve less invasive treatment—only a bath of heated solution. They can be 3D-printed with polyvinyl alcohol (PVA) filaments, a water-soluble material typically used as support material in fusion deposition modelling (FDM) [27,30]. In [34], a water-soluble resin w used to 3D print through stereolithography (SLA) the inner mould of brain vasculature phantoms and obtain a flow model in the form of a transparent silicone box.

In this paper, we present a manufacturing process to obtain compliant and transparent vascular phantoms for in vitro validation of computational simulations (structural and fluid dynamics) of CA endovascular treatment. The workflow consisted of a dip–spin silicone coating technique of 3D-printed water-soluble inner moulds. We compared two printing techniques to produce the inner moulds: fusion deposition modelling and stereolithography. To the authors’ knowledge, this work represents one of the first applications of water-soluble resins for moulding patient-specific brain arteries. Since these models were designed for in vitro experiments and the validation of in silico simulations, extensive characterisation was carried out in terms of the mechanical properties of the silicone and the thickness and transparency of the final models. The remainder of the paper is organised as follows: in Section 2, the process is described with a detailed description of all fabrication passages; in Section 3, the final models, the results of the silicone mechanical characterisation, and the transparency and thickness uniformity tests are presented and discussed; finally, in Section 4, some concluding remarks are made.

## 2. Material and Methods

The fabrication process consisted of five steps (Figure 1):
Build the CAD model of the vessel surface (idealised model) or obtain the 3D surface of the vessel from medical images through segmentation (patient-specific model);Obtain the inner mould of the vessel surface by 3D printing a water-soluble material. Two printing techniques, stereolithography and fused-deposition modelling, were compared for the production of the water-soluble inner mould;Post-process the 3D-printed mould;Manually encase the inner mould in transparent silicone using a spin–dip coating technique;Dissolve the water-soluble mould and extract the silicone model.

Each step is described in detail in the following sections.

### 2.1. Vessel Design

Two different vessel designs were considered in this study:Idealised models of a cerebral internal carotid artery characterised by the presence of a saccular aneurysm (Figure 2A);Realistic models whose geometries were selected from those available in the free AneuriskWeb dataset repository [35] (Figure 2B).

The average diameter of the internal carotid is 4.66 mm for women and 5.11 mm for men [36]. Smaller and larger diameters were also considered in this work to assess the final spatial resolution of the proposed manufacturing process. Thus, idealised vessels with a diameter (*D*_vessel_) ranging between 2 and 6 mm were modelled. Compared with solid (filled) ones, hollow inner moulds increase the surface area in contact with water and accelerate the dissolution process; therefore, a negative offset (*s*) of 0.5/0.75 mm was added to the vessel surface. Patient-specific geometries were adjusted by adding a tubular extension to one of the extremities to accommodate the axis of rotation used during the curing step.

### 2.2. 3D Printing

The water-soluble mould of the vessel surface was 3D-printed using two different techniques:Fused-deposition modelling using a PVA-based filament (Figure 3A);Stereolithography using a water-soluble resin (Figure 3B).

A brief explanation of both is detailed in the next chapters.

#### 2.2.1. Fusion Deposition Modelling

In FDM, printing is achieved by a thin filament of a thermoplastic material that is deposited on the printing bed layer by layer by a computer-controlled extruder. In this work, the Meccatronicore Studio 300 3D printer (Meccatronicore S.R.L., Pergine Valsugana, TN, Italy) and the GhostIce filament (Paperdifferent, Marghera, VE, Italy) were used. This PVA-based filament, contrary to the standard one, was chosen since it leaves fewer slobbery residues on the final surface. Its printing settings are reported in Table 1. The 3D printer employed had a nozzle diameter of 0.4 mm. Particular attention was paid to the percentage of infill of the 3D-printed models. Greater infill results in a more rigid and resistant model, which is an advantage for subsequent manipulations to which the model is subjected; however, it also prolongs the dissolution process in water. A good compromise was found with 85% infill for larger models, while 100% infill was necessary for smaller models (*D*_vessel_ < 3mm) to prevent them from being too weak and breaking during later manipulations.

#### 2.2.2. Stereolithography

In SLA, a computer-controlled laser beam is focused onto a photo-polymerisable resin, and the full model is obtained by curing this liquid layer by layer. In this work, the 3D printer employed was a Zortrax Inkspire (Zortrax, Olsztyn, Poland), while a water-soluble resin was selected for this study. Its printing settings are reported in Table 2. The models printed by SLA needed to be UV-cured for 20 min at 35 °C before manipulations. The 3D printer employed had an XY resolution of 50 microns and a minimum layer thickness of 25 microns.

### 2.3. Post-Processing

The 3D-printed inner moulds underwent a series of successive treatments before coating. The base and supports were carefully removed with a scalpel. A polishing step was performed to reduce rough spots left by the supports on the surface (for SLA and FDM models) and to reduce the overall surface roughness (for FDM models). The operation was performed with fine and extra-fine sandpaper to avoid compromising the quality of the model (Figure 4A,B). Finally, we liquefied the solid resin and filaments left over from the removal of the supports and used the resulting liquid to paint the surface of the moulds and counterbalance the material removal due to polishing (Figure 4C). This was essential to avoid the formation of air bubbles during the next step.

### 2.4. Coating

Among those available, the two-component silicone rubber KE-1603-A (Shin-Etsu Chemical Co., Ltd., Chiyoda Tokyo, Japan) was deemed the most suitable coating material in terms of transparency, elasticity, and tear resistance. The characteristics provided by the vendor are reported in Table 3. Because of the partial information, the variations that may have been present due to curing time and temperature, and the need to have the entire stress–strain curve to compare in silico and in vitro simulations, uniaxial tensile tests were performed on dog bone samples. These tests represent the golden standard to investigate the mechanical behaviour of isotropic materials. The corresponding results are reported in Section 3.3.

The coating was accomplished by dipping the model into the silicone mixture; a brush-painting technique was previously attempted, but the high viscosity of the silicone together with the manual application resulted in the encapsulation of too many bubbles. To facilitate dipping and spinning, the 3D-printed mould was fixed on a straight stick (Figure 4C). Once the silicone components were mixed and the bubbles were removed by a vacuum pump; the transparent layer was applied by dipping the 3D-printed mould into the silicone mixture. The model was taken out, and excess material and bubbles were removed manually. To achieve an even distribution of the silicone layer, the model was placed on a rotating axis; the rotation was achieved with a stepper motor whose speed was controlled via a single-board microcontroller (Arduino Uno Rev. 3; Arduino SRL; Italy). The setup is shown in Figure 5. The rotation speed was set to 10 RPM. Finally, the rotating coated model was placed in a temperature-controlled environment at 30° to harden the silicone. Higher temperatures may accelerate solidification, but they also increase the possibility of microbubble formation. To ensure that the silicone solidified, a curing time of at least 4 h was considered.

### 2.5. Dissolving Procedure

When the transparent layer solidified (Figure 4D), the inner mould was dissolved in a hot water bath using a magnetic stirrer at 90 °C. Manipulations were performed to accelerate the removal of the inner mould (Figure 4E).

## 3. Results and Discussion

### 3.1. Inner Mould: 3D Printing Technique Comparison

The inner moulds obtained through FDM required a printing time varying from 30 to 45; by contrast, SLA printing corresponded to printing times larger than 5 h.

The FDM-printed inner moulds had more irregular surfaces than those obtained through SLA (Figure 6A,B); in fact, polishing was restrained so as not to compromise the resolution of the printed model, and thus, surface imperfections due to layer deposition were still visible in the FDM-printed moulds. This difference was related to the resolution of the two printing techniques; our SLA 3D printer presented an XY resolution that was 8 times higher than the nozzle diameter of the FDM printer. This was reflected in the quality of the final results: the SLA-printed models incorporated fewer microbubbles and showed a smoother inner wall (Figure 6C,D).

Using FDM, the minimum attainable diameter was 2 mm. For smaller diameters, the 3D-printed models became too fragile to support the silicone layer and bent due to the silicone weight while curing in the heating chamber. For SLA, it was not always possible to guarantee the hollowness of models with a diameter of 3 mm; therefore, we stopped the comparison at this value. We believe that this was entirely related to the poor performance of the water-soluble resin and not to the 3D printing technique. The water-soluble resin was found to be inconsistent in terms of printed results, with the inner mould exhibiting mechanical properties strongly affected by storage conditions (humidity and temperature) and becoming fragile to handle and thus difficult to manipulate for support removal. Moreover, non-hollow models implied longer dissolution times: 5/6 h compared with 1.5 h for FDM-printed models. With some complex geometries, the complete dissolution of SLA-printed moulds was never reached, even after a long bath period (Figure 6B).

### 3.2. Final Models: Evaluation

As visible in Figure 7, the patient-specific anatomical replicas qualitatively respect the original surface model, regardless of the 3D printing technique employed to manufacture the inner mould. The main coating defects are present in the form of silicone accumulation in the connecting sections (transition between different curvatures and aneurysm neck) and the microbubble encapsulation. These defects are highlighted in Figure 8 for idealised models obtained from FDM-printed moulds, but the same applies to those obtained through SLA. Silicone accumulation results in uneven thickness and, therefore, inhomogeneous mechanical properties of the final model. This can be explained by the use of high-density silicone and the simple rotation system performed only around a horizontal axis.

### 3.3. Silicone Layer: Material Characterisation

Dog bone samples of silicone rubber KE-1603-A were obtained by die cutting to ensure uniform thickness. They had a gauge length of 33 mm and a rectangular cross-section thickness of 4 × 1 mm. The thickness of the samples was close to that of the final models. Uniaxial tensile tests were performed until rupture on five samples in air at room temperature using an MTS Insight 5 (MTS Systems Corporation, Eden Prairie, MN, USA) with a loadcell of 5 kN. The extension rate was set to a constant speed of 500 mm/min. The strain was evaluated by dividing the stroke reading by the gauge length. The measured data were filtered using a moving mean with a sliding window with a length of 20. The elastic modulus was computed as the slope of the stress–strain curve. Although the results varied significantly among samples, the average tensile strength and elongation at break were close to those indicated by the vendor (Table 3). The difference in tensile strength at break could be explained by the presence of defects on the side of the samples due to die cutting. In any case, we were interested in the elastic part of the curve since our models were designed to work with small deformations: this part of the curve matched very well for all samples. From the results shown in Figure 9, it can be noticed that this silicone presented non-linear elastic behaviour with an elastic modulus that increased from 0.6 to 1.6 MPa for strains between 0.5 and 2 and decreased to 0.5 MPa before rupture. The measured Young’s modulus showed good agreement with that described in the literature from in vitro testing of CA specimens. As a comparison, some examples are reported in Table 4.

### 3.4. Final Model: Thickness Quantification

The thickness of the silicone vascular models was measured to validate the proposed method. Two idealised models with different internal diameters were considered: *D*_vessel_ = 5 mm for the first model (A) and *D*_vessel_ = 3 mm for the second model (B). Both models were obtained using FDM-printed inner moulds. The local thickness of the models was measured with a micrometre (Palmer’s type) at four circumferential points in seven sections along the longitudinal direction to monitor the thickness variation. The measurements were taken from cross-sections vertical to the substrate. Figure 10 shows the thickness data of the two models. The average thickness was 1.0317 ± 0.21 mm for the large-diameter model and 1.0299 ± 0.14 mm for the small-diameter model. The highest variance could be observed near the connecting sections, particularly at the edges of the aneurysm, while the dome of the aneurysm was thinner than the tube section. Therefore, it can be said that the thickness uniformity requirements were mostly met in the longitudinal direction, while in the vertical direction (within sections), there is still room for improvement. As observed by others in previous studies [28], smaller models were less affected by thickness inhomogeneities.

### 3.5. Final Model: Transparency Evaluation

To validate the transparency of the final models, a simple RI matching test was conducted to analyse the distortion due to the presence of the model on a background grid as the working fluid changed. The test was repeated in air (RI = 1), water (RI = 1.33), and a 50/50 glycerol/water solution. Glycerol has a higher RI than water and dissolves completely in the latter.

We compared the transparency of three models:A single-silicone-layer model obtained with the workflow proposed here (S1);A triple-silicone-layer model obtained with the workflow proposed here (S2);A model printed using a transparent rigid resin (S3). The resin used was Somos WaterClear Ultra 10122 (Covestro Additive Manufacturing, Geleen, The Netherlands) and was 3D-printed using an SLA 3500 3D printer (3D Systems, Rock Hill, SC, USA).

The purpose of this last model was to compare transparent rigid models obtained by SLA with the silicone models obtained by the indirect 3D printing method proposed here and verify whether silicone had a lower refractive index than transparent resins.

The results are reported in Figure 11. When the models were placed in air or water, the background grid was visible but distorted. When glycerol was added to water and thus the RI of the working fluid increased, the optical distortion on the lines was decreased; S1 and S2 disappeared completely in a 50/50 glycerol/water solution. It follows that these models had an RI approximately equal to 1.4. Since no noticeable differences were observed between S1 and S2, it can be stated that the transparency properties of KE-1603-A silicone rubber are guaranteed even with thicker models. By contrast, S3 remained visible in the glycerol–water mixture even at a higher glycerol percentage, proving that transparent rigid resins have a high RI, which is difficult to match with common working fluids.

## 4. Conclusions

In this paper, we proposed a workflow for manufacturing hollow elastic models for in vitro studies of CAs. Two techniques for obtaining water-soluble inner moulds were investigated; the FDM method was found to be more appropriate given the reliability of the final results and the facility to perform the superficial post-processing to remove supports and smooth the surface of the inner mould. Further investigations on water-soluble resins for SLA will be considered in the future. From the mechanical tests on the silicone samples, we observed nonlinear elastic behaviour, which could not be inferred from the information provided by the vendor. We found that the measured Young’s modulus was in line with that described in the literature for in vitro testing of brain vessels and aneurysms. This allowed us to obtain more realistic values for the wall material properties to be used for the comparison between in vitro and in silico simulations. The vessel models matched the refractive index of a 50/50 glycerol/water solution, which makes them suitable for application in optical flow measurement techniques. The main limitation of this workflow is the need for several manual operations, whose results are intrinsically operator-dependent; a potential solution would be to automatise the dipping process and use non-manual polishing techniques, such as a fluidised bed. Moreover, future technical developments will include the implementation of a pseudo-random rotation system to improve the thickness uniformity of the model. Further characterisation of the models will involve a quantitative comparison of the final shape with the original one through CT scans and the measurement of the friction coefficient of the resulting silicone models.

## Figures and Tables

**Figure 1 materials-16-00166-f001:**
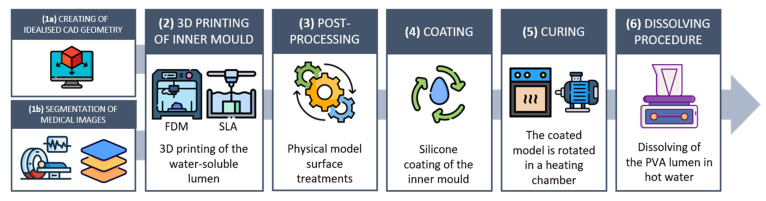
The proposed manufacturing process: (**1a**) acquisition of patient-specific clinical images and segmentation to reconstruct the vasculature geometry; (**1b**) creation of the CAD simplified geometry; (**2**) 3D printing of the vessel inner mould in water-soluble material; (**3**) physical model refinement (removal of 3D printing supports, polishing); (**4**) spin–dip coating and (**5**) curing; and (**6**) dissolving procedure in a hot water bath.

**Figure 2 materials-16-00166-f002:**
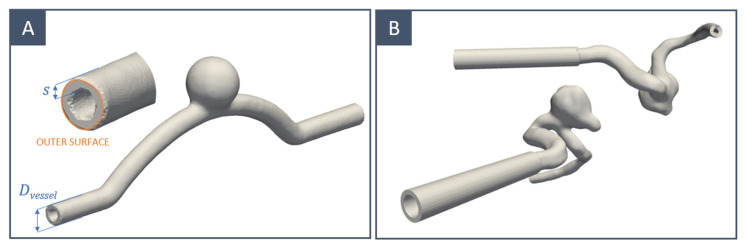
CAD geometry with offset for the inner mould of (**A**) idealised and (**B**) patient-specific geometry.

**Figure 3 materials-16-00166-f003:**
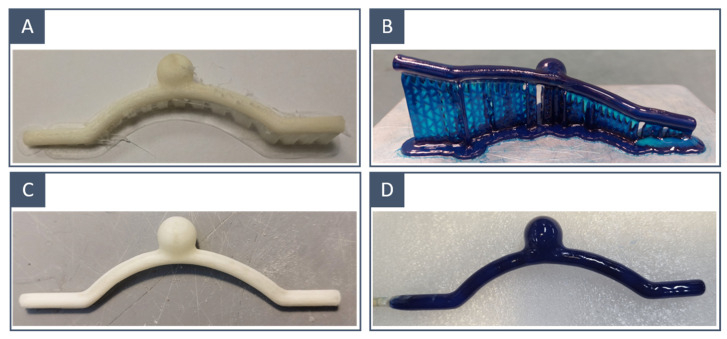
The 3D-printed inner moulds before and after surface treatment: FDM-printed idealised model (**A**) before and (**C**) after postprocessing; SLA-printed idealised model (**B**) before and (**D**) after postprocessing.

**Figure 4 materials-16-00166-f004:**
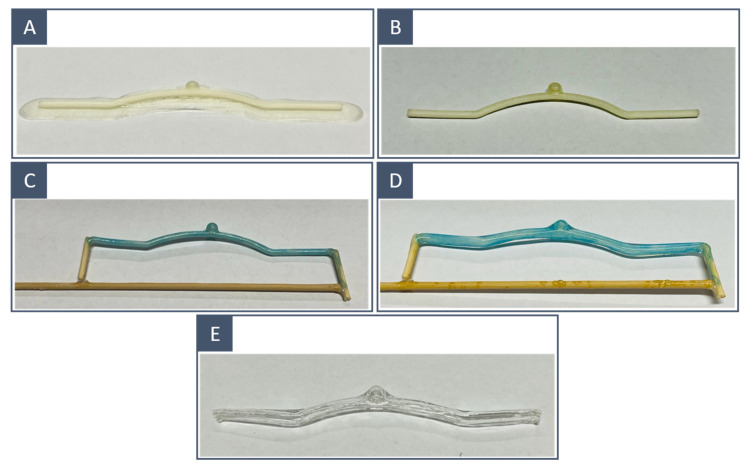
Manufacturing process: (**A**) FDM-printed model with supports, (**B**) after supports removal and surface polishing, (**C**) painting with the liquified PVA-based filament and water-soluble resin solution and fixing on a straight stick, (**D**) after solidification of the silicone layer and (**E**) after dissolution. The same workflow was carried out on SLA-printed models.

**Figure 5 materials-16-00166-f005:**
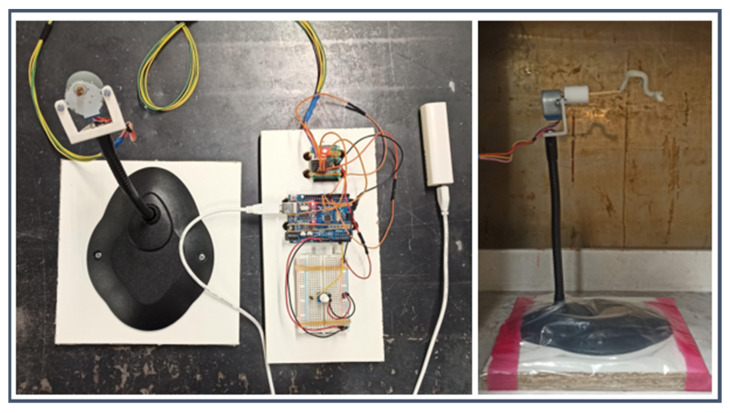
Set-up to rotate the models inside the heating chamber.

**Figure 6 materials-16-00166-f006:**
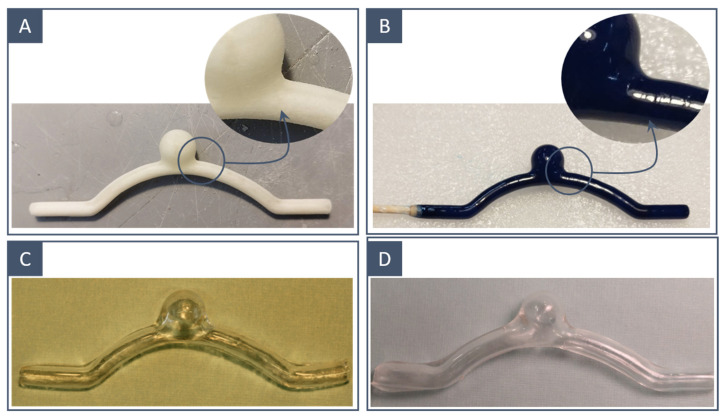
Comparison between 3D printing techniques to obtain water-soluble inner moulds: (**A**) FDM-printed inner mould and (**B**) SLA-printed inner mould; (**C**) final model corresponding to FDM-printed inner mould and (**D**) final model corresponding to SLA-printed inner mould.

**Figure 7 materials-16-00166-f007:**
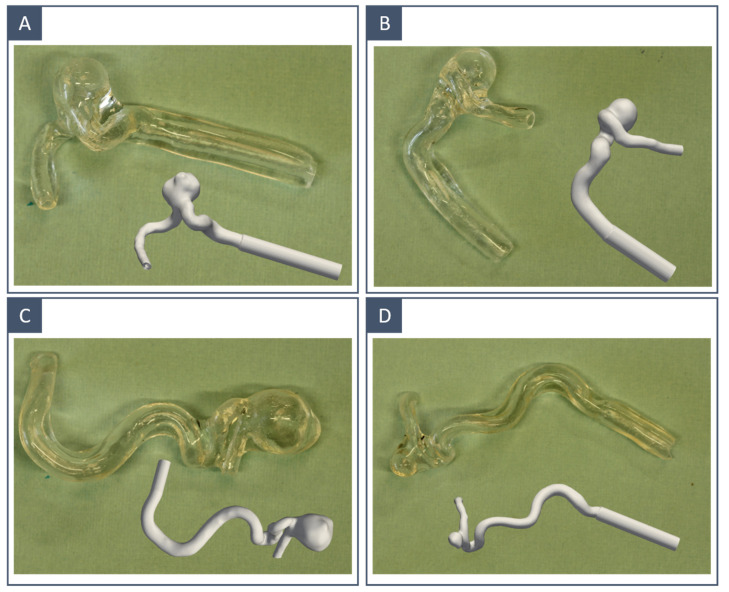
Patient-specific phantoms with the corresponding 3D surface model. The surface models were selected from those available in the free AneuriskWeb dataset repository. Models (**A**,**B**) were obtained with an FDM-printed inner mould; models (**C**,**D**) were obtained with an SLA-printed inner mould.

**Figure 8 materials-16-00166-f008:**
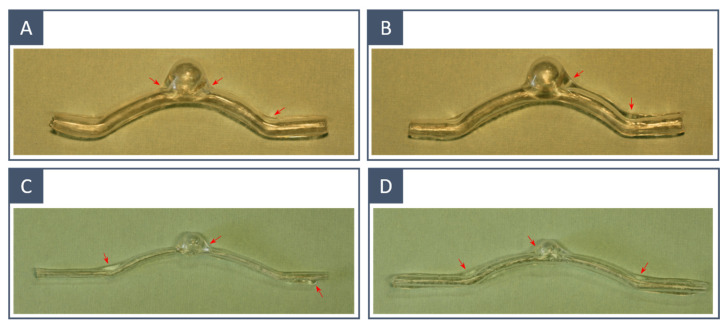
Idealised phantoms obtained from FDM-printed moulds: the main fabrication defects are highlighted with red arrows. Models (**A**,**B**) are two examples of large vessels (*D*_vessel_ = 5 mm); models (**C**,**D**) are examples of small vessels (*D*_vessel_ = 3 mm).

**Figure 9 materials-16-00166-f009:**
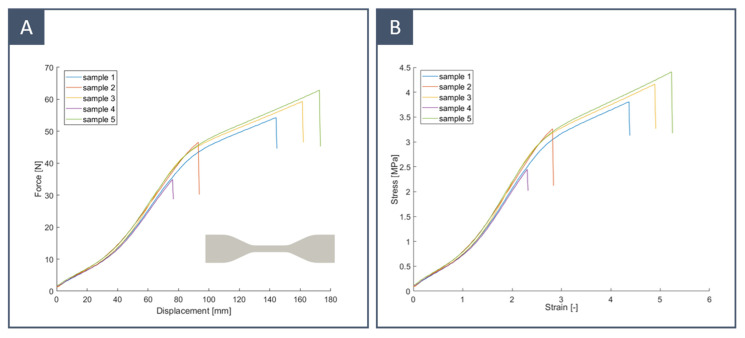
Uniaxial tests for five dog bone samples of silicone rubber KE-1603-A: (**A**) force–displacement curve; (**B**) stress–strain curve.

**Figure 10 materials-16-00166-f010:**
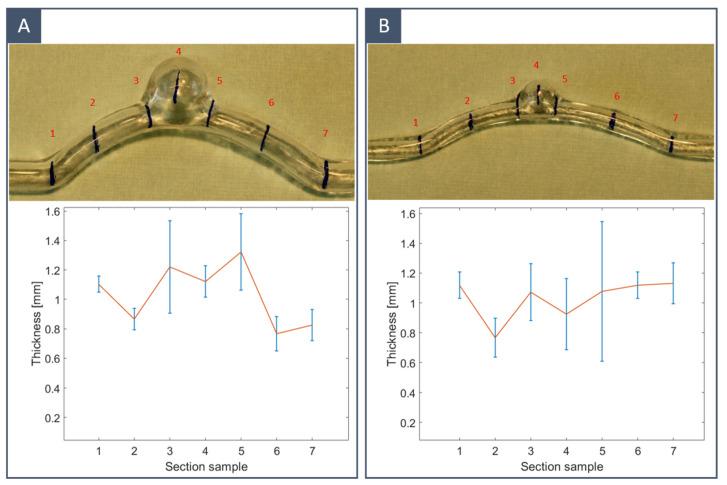
Average thickness (red line) and standard deviation (blue bars) for two idealised models: (**A**) *D*_vessel_ = 5 mm and (**B**) *D*_vessel_ = 3 mm. The local thickness of the models was measured with a micrometre (Palmer’s type) at four circumferential points in seven sections (labelled with red numbers) along the longitudinal direction.

**Figure 11 materials-16-00166-f011:**
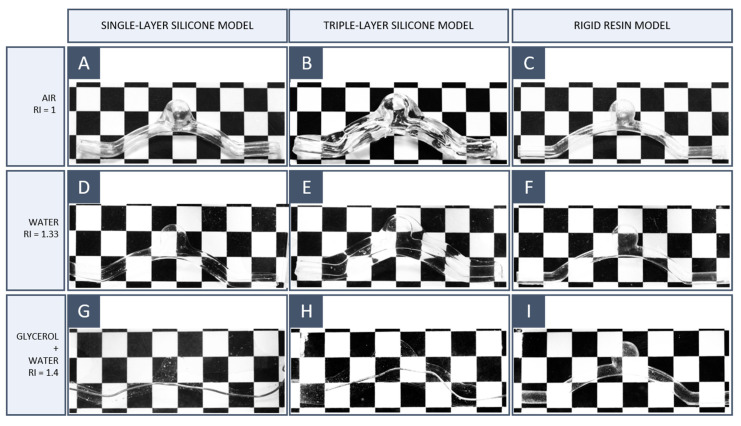
Index of refraction matching test for 3D printing model: (**A**–**C**) air, (**D**–**F**) water, and (**G**–**I**) 50/50 glycerol/water solution. The first column corresponds to the single silicone layer model (S1); the second corresponds to the triple silicone layer model (S2) and the third corresponds to the rigid SLA resin model (S3).

**Table 1 materials-16-00166-t001:** FDM printing settings for GhostIce (Paperdifferent, Marghera, VE, Italy).

Layer Height [mm]	Printing Temperature [°C]	Build Plate Temperature [°C]	Print Speed [mm/s]
0.075	194	70	70

**Table 2 materials-16-00166-t002:** SLA printing settings for water soluble printable resin.

Layer Height [mm]	Layer Exposure Time [s]	Bottom Layers Exposure Time [s]	Bottom Layers [–]	Exposure Off Time [s]	z-Lift Distance [mm]	Platform Lift Distance [mm/min]
0.05	20	170	5	2	5	100

**Table 3 materials-16-00166-t003:** Two-component silicone rubber KE-1603-A after-curing characteristics.

Density 23 °C [g/cm^3]^	Hardness Durometer A	Tensile Strength [MPa]	Elongation at Break [%]	Tear Strength [kN/m]
1.03	28	3.5	450	12

**Table 4 materials-16-00166-t004:** Average mechanical data from in vitro studies on CA specimens evaluated by uniaxial strain/stress measurements [37].

Structure	Elastic Modulus MPa
Aneurysm fundi	1.7 ± 0.8
Aneurysm necks	3.1 ± 0.9
Arteries	2.5 ± 1.1

## Data Availability

The surface models were selected from those available in the free AneuriskWeb dataset repository [35].

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
