# Peer review of "Fabrication of Compliant and Transparent Hollow Cerebral Vascular Phantoms for In Vitro Studies Using 3D Printing and Spin–Dip Coating"

_materials, 2022, doi:10.3390/ma16010166_

Round 1

Reviewer 1 Report

The authors suggested a manufacturing process to obtain compliant and transparent hollow vessel replicas to assess the mechanical behavior of endovascular devices and perform flow measurement. 

1. I can not find the originality of this research in the introduction section. It seems that everything has been done in the previous studies in the literature review part (inner mould is 3d printed, coated in rotating conditions, etc). What does "easily implementable manufacturing process" mean compared to the previous research?

2. Although there are two different 3d cad models of inner mold are suggested, there are no comparison results for those models. It seems that the specimen in Fig. 3 are printed from only an idealized model. If both models are not analyzed, there is no reason to explain for the model (surface model with offset in Fig. 3(b)

3. Could you explain that the water-soluble resin used for SLA printing in section 2.2 is the same as the one mentioned in section 3.4? It is so confusing because there are no relevant analysis results on how the final model is obtained. 

4. In section 2.3, are there any results of SLA-printed model?

5. The explanation of Fig. 5 in section 2.4 is confusing. 

 - "the model is taken out and excess material and bubbles are removed manually". It doesn't match Fig. 5(a). Why is the upper picture in Fig. 5(a) added as it has a different shape from the model in Fig. 2. 

 - It is the same for Fig. 5(b), which is more look like test setup. 

6. How did the authors come up with the final model in the result section? Which 3d printing technology is used for the final model? Also, I don't understand why Fig. 6 is added. 

7. Authors explained that SLA-printed models show smoother surfaces in Fig. 8. However, it is difficult if the surface is smoother than the one in the FDM-printed model from the picture. There should be some measurement data or visual data for comparison. 

8. It seems that the material is more important than printing technology. Normally, SLA is well known for its higher precision. However, the authors suggested that FDM is more appropriate for this research. 

9. Why the uniaxial test data is added in the manuscript? I don't see any reason these results are important because there is no proper explanation. 

10. In Fig. 11, it is also confusing that which printing method is used for the sample and if it is coated or directly printed. 

11. It is difficult to find academic achievement from the results. 

Author Response

The authors suggested a manufacturing process to obtain compliant and transparent hollow vessel replicas to assess the mechanical behavior of endovascular devices and perform flow measurement. 

  1. I can not find the originality of this research in the introduction section. It seems that everything has been done in the previous studies in the literature review part (inner mould is 3d printed, coated in rotating conditions, etc). What does "easily implementable manufacturing process" mean compared to the previous research?

We believe that the work presented in this article differs from those already in the literature because we compared two different techniques for obtaining internal moulds and because we performed an extensive characterization of the final models in view of their use for in-vitro studies. Regarding the printing techniques, the work represents one of the first applications of water-soluble resins for moulding patient-specific cerebral arteries. Through their characterization, we ensured that the final models can be used for in-vitro optical studies: in fact, their mechanical properties are close to those measured from in-vitro testing of cerebral aneurysm specimens and their refractive index corresponds to a commonly used working fluid (glycerol+water). Future developments are certainly planned to improve the quality of the models, but the models obtained with the described workflow are already suitable for the envisaged application. We added some lines in the Introduction to better clarify this point.

  1. Although there are two different 3d cad models of inner mould are suggested, there are no comparison results for those models. It seems that the specimen in Fig. 3 are printed from only an idealized model. If both models are not analysed, there is no reason to explain for the model (surface model with offset in Fig. 3(b))

The two different CAD models (with or without offset) are not intended for comparison: if 3D printed, CAD models without offset results in a solid, filled model that would take longer to dissolve. Therefore, CAD models with offset are used since hollow models increase the surface area in contact with water and accelerate the dissolution process. Since this was unclear, we changed Figure 2 and left only examples of models with offset (one idealised and one patient-specific).

  1. Could you explain that the water-soluble resin used for SLA printing in section 2.2 is the same as the one mentioned in section 3.4? It is so confusing because there are no relevant analysis results on how the final model is obtained. 

We did not mention any SLA-printed model in Section 3.4; perhaps the reviewer referred to Section 3.5. In fact, in this section, we mentioned an SLA-printed model that is not the inner SLA-printed water-soluble mould mentioned in Section 2.2. The SLA printed model mentioned here is a transparent rigid phantom directly 3D printed with a transparent resin (WaterClear Ultra 10122 by Covestro Additive Manufacturing, Netherlands). Its purpose is to compare transparent rigid models obtained with SLA with silicone models obtained with the indirect 3D method proposed in our work and to show that silicones have a lower refractive index than transparent resins. According to the reviewer’s comment, we clarified this aspect in Section 3.5.

  1. In section 2.3, are there any results of SLA-printed model?

Yes, the same post-processing technique is performed for SLA-printed and FDM-printed models. FDM models require polishing not only to reduce the spots left by supports but also to reduce the surface roughness due to the 3D printing technique itself; on the other hand, the removal of support traces on SLA-printed models requires more time and labour. We have left the figure with the workflow only for the FDM-printed model, but we have specified in the text and description of the figure that the same workflow is performed for SLA-printed models.

  1. The explanation of Fig. 5 in section 2.4 is confusing. 
    1. "the model is taken out and excess material and bubbles are removed manually". It doesn't match Fig. 5(a). Why is the upper picture in Fig. 5(a) added as it has a different shape from the model in Fig. 2. 

These images were intended to show two different inner moulds after coating.

  1. It is the same for Fig. 5(b), which is more look like test setup. 

The figure represents the setup used to rotate the models inside the heating chamber. 

Based on the reviewer's comment, we modified Figure 5: the entire production process is now shown in Figure 4, and in Figure 5 we left only the set-up used to rotate the models inside the heating chamber.

  1. How did the authors come up with the final model in the result section? Which 3d printing technology is used for the final model? Also, I don't understand why Fig. 6 is added. 

Based on the reviewer's comment, we changed the order of the sections: we first compare the results of the two 3D printing techniques considered (Section 3.1: INNER MOULD: 3D PRINTING TECHNIQUE COMPARISON) and then, evaluate from a quality point of view, whether the final models match the original CAD models and what are the main sources of error, regardless of the printing technique (3.2: FINAL MODELS; EVALUATION). We also clarified which 3D printing technology was used for each of the models shown in the images. Figure 6 refers to a series of patient-specific phantoms obtained with both 3D printing techniques: it is intended to show the ability of our workflow to work not only with idealized models but also with more complex, patient-specific models.

  1. Authors explained that SLA-printed models show smoother surfaces in Fig. 8. However, it is difficult if the surface is smoother than the one in the FDM-printed model from the picture. There should be some measurement data or visual data for comparison. 

We changed the former Figure 8 (current Figure 6) to visually compare the surface of FDM and SLA-printed inner moulds and the related final models as suggested by the reviewer.

  1. It seems that the material is more important than printing technology. Normally, SLA is well known for its higher precision. However, the authors suggested that FDM is more appropriate for this research. 

This is indeed the main conclusion we drew from our work. The SLA technique allows for more accurate and smoother 3D surfaces thanks to the higher printing resolution compared to FDM; however, the water-soluble resin we tested (one of the few available on the market) is inconsistent in terms of the printing results, with the inner mould exhibiting mechanical properties that are strongly affected by storage conditions (humidity, temperature) and becoming brittle to handle and therefore, difficult to manipulate for supports removal. In addition, in most 3D printed models with SLA, we were unable to guarantee hollowness, which makes the dissolution process take longer or never be fully achieved. FDM models are therefore deemed more appropriate for this research. As stated in the conclusions, further investigation of water-soluble resins for SLA will be considered in the future.

  1. Why the uniaxial test data is added in the manuscript? I don't see any reason these results are important because there is no proper explanation. 

Data from uniaxial tests were added because of the partial information provided by the vendor on the material properties of the silicone used, the variations that may be present due to curing time and temperature and the need to have the entire stress-strain curve for future comparisons between in-silico and in-vitro simulations. Moreover, this allowed us to obtain realistic values for the wall material properties, which were compared with those present in the literature from in vitro-tests on brain aneurysms showing a good agreement. We have added clarification on this in Sections 2.4, 3.3 and 4.

  1. In Fig. 11, it is also confusing that which printing method is used for the sample and if it is coated or directly printed. 

The image was fixed according to the reviewer’s comment and substituted.

  1. It is difficult to find academic achievement from the results. 

This work includes extensive characterization of the final models, in terms of quantification of thickness, transparency and mechanical behaviour of the material. We were unable to find similar works in the literature: typically, only one or two such characterizations are presented. Thus, we were able to ensure that our models can be used for physiologically representative in vitro tests. Furthermore, our work stands out because we compared the use of two different printing techniques (showing the advantages and disadvantages of both) and propose the use of a commercial silicone, with excellent transparency properties (even at higher thicknesses), never before used in literature.

Reviewer 2 Report

For the manuscript “Fabrication of compliant and transparent hollow cerebral vascular phantoms for in-vitro studies using 3D printing and spin-dip coating”, the authors prepared compliant and transparent hollow cerebral vascular phantoms by 3D printing technologies, and the relevant tests were carried out. The research has certain innovative and application value. In my opinion, major revisions are needed before it can be published in “materials, and the detailed comments are as follows:

1. The introduction states " To date, techniques to build compliant vessel models can be traced to two main categories: direct and indirect additive manufacturing (AM). The former is explained in the text, so the latter would better be classified and described in a relevant way.

2. Furthermore, the states "The former is based solely on Polyjet 3D printing". I think this statement is not rigorous enough.

3. Printing accuracy is an important parameter in the 3D printing process. FDM and SLA processes are chosen for 3D printing in this article, but the printing accuracy of the related technologies is not described, please add the relevant content.

4. There is a lot of blank space on page 6, please adjust.

5. This paper describes " polished using sandpaper" in section 2.3. I wonder if polished with sandpaper would be so uncontrollable that it would affect the accuracy and performance of the structure itself?

6. This paper adopts "manual operation" for several experimental steps, and I would like to know whether there will be too much error in the manual operation that will affect the accuracy of the printed objects.

Author Response

For the manuscript “Fabrication of compliant and transparent hollow cerebral vascular phantoms for in-vitro studies using 3D printing and spin-dip coating”, the authors prepared compliant and transparent hollow cerebral vascular phantoms by 3D printing technologies, and the relevant tests were carried out. The research has certain innovative and application value. In my opinion, major revisions are needed before it can be published in “materials”, and the detailed comments are as follows:

  1. The introduction states " To date, techniques to build compliant vessel models can be traced to two main categories: direct and indirect additive manufacturing (AM). The former is explained in the text, so the latter would better be classified and described in a relevant way.

We dedicated the entire paragraph from line 60 to line 74 to the description of indirect addictive manufacturing methods.

  1. Furthermore, the states "The former is based solely on Polyjet 3D printing". I think this statement is not rigorous enough.

During our literature research, no other direct 3D printing technique other than Polyjet 3D printing was found that allows to obtain compliant models with tissue-like properties. However, according to the reviewer, a more general statement has been replaced.

  1. Printing accuracy is an important parameter in the 3D printing process. FDM and SLA processes are chosen for 3D printing in this article, but the printing accuracy of the related technologies is not described, please add the relevant content.

FDM-printed inner moulds have a more irregular surface than those obtained through SLA: in fact, polishing is restrained so as not to compromise the resolution of the printed model and, thus, surface imperfections due to layer deposition are still visible in the FDM-printed moulds. This difference is related to the resolution of the two printing techniques: the used 3D printer for SLA present a XY solution which is 8 times higher than the nozzle diameter of the FDM printer. This is reflected in the quality of the final results: SLA-printed models incorporate fewer microbubbles and show a smoother inner wall. We added this description in Section 3.1 and highlighted the difference in the related figure (former 3.2).

  1. There is a lot of blank space on page 6, please adjust.

This will be fixed in the final version.

  1. This paper describes " polished using sandpaper" in section 2.3. I wonder if polished with sandpaper would be so uncontrollable that it would affect the accuracy and performance of the structure itself?

We minimized manual intervention as much as possible but polishing is needed to reduce the traces left by the supports on the surface (for SLA and FDM models) and to smooth the surface for FDM models. The operation was performed with fine and extra-fine sandpaper to avoid compromising the quality of the model. Moreover, we liquefied the resin and filaments left over from the removal of the supports and used the resulting liquid to counterbalance the polishing step. We clarified this aspect in Section 2.3.

  1. This paper adopts "manual operation" for several experimental steps, and I would like to know whether there will be too much error in the manual operation that will affect the accuracy of the printed objects.

We believe that the main errors that could be introduced manually are related to polishing (as explained above) and silicone deposition (too much silicone can lead to the formation of silicone accumulation, while too little silicone can lead to the formation of holes). This makes the process very dependent on personal experience. For this reason, we propose in the future to automatize also the dipping process and the use of non-manual polishing techniques, such as a fluidized bed. We clarified this point in the Conclusions.

Reviewer 3 Report

This publication described a technique for creating hollow cerebral vascular phantoms utilizing a spin-dip coating and 3D-printed water-soluble molds. The study is properly described. Before publication, there are a few problems that need to be clarified. In the introduction section, a review of the literature on current 3D printing technologies for creating vascularized models is provided. However, it is not immediately obvious why a low refractive index is preferred for in vitro vascular models. If the author could give examples of or references to any optical experiments used in cerebral aneurysm investigations, that would be great. In Section 2.4 Coating, what's the rotation speed for the stepper motor? And how will the spin settings influence the coating layer's consistency? In Section 3.2, can you describe on the printing resolution and final accuracy compared with design specifications between the two printing methods? In addition, polishing was added as a post-care procedure for FDM-manufactured molds. How will this affect the surface quality and overall accuracy of the printed item in comparison to the CAD design? And what is the smallest feature size that can be produced using either printing technique in terms of the inner diameter of the finished tubular structure? For Section 3.3, can you provide some insight into the variation between different sample parts observed in the strain-stress curve? is it because of material inconsistency or manufacturing defects in the dog-bone samples? Given the silicone was a commercially available material, how does the outcome compare to the material attributes the vendor stated? Furthermore, is the material property as determined from a bulk material a reliable indicator of the tubular thin-wall structure? For Section 3.4, based on the marker in the photo, it is unclear whether measurements were taken from a cross-section that was vertical to the substrate plane or perpendicular to the local axis. For figure 11, it would be preferable to include the RI results for each figure as a comparison. Additionally, a fluid test that compares the suggested vascular phantoms to in vivo models can be used to show how well they work fluidically. I'm asking because a tubular structure's fluid mechanics may also be impacted by the surface hydrophilic characteristic. 

Author Response

This publication described a technique for creating hollow cerebral vascular phantoms utilizing a spin-dip coating and 3D-printed water-soluble molds. The study is properly described. Before publication, there are a few problems that need to be clarified.

  1. In the introduction section, a review of the literature on current 3D printing technologies for creating vascularized models is provided. However, it is not immediately obvious why a low refractive index is preferred for in vitro vascular models. If the author could give examples of or references to any optical experiments used in cerebral aneurysm investigations, that would be great.

Optical methods for flow visualization and measurement consist in seeding the fluid with small tracer particles, illuminating them with laser pulses, taking subsequent pictures of the illuminated particles with a camera and, finally, reconstructing their motion. These techniques have the advantage of not disturbing the flow being studied. If the model in which the fluid is flowing presents a curved surface, the light signal is distorted and this leads to errors in the reconstructed field. If the surface is complex, no easy post-processing correction can be applied. The most typically exploited solution is refractive-index matching, where the idea is to choose a working fluid and a model material whose refractive indices are very close. We rephrased what was written before and added a better clarification about it in the Introduction.

  1. In Section 2.4 Coating, what's the rotation speed for the stepper motor? And how will the spin settings influence the coating layer's consistency?

We set the rotation speed to 10 RPM. No influence on the silicone consistency was found; however, if the silicone is too liquid, it will drop while solidifying so higher velocities are preferred. We added a sentence on the rotation speed in Section 2.4.

  1. In Section 3.2, can you describe on the printing resolution and final accuracy compared with design specifications between the two printing methods?

FDM-printed inner moulds have a more irregular surface than those obtained through SLA: in fact, polishing is restrained so as not to compromise the resolution and, thus, surface imperfections due to layer deposition are still visible in the FDM-printed moulds. This difference is related to the resolution of the two printing techniques: the used 3D printer for SLA presents a XY solution which is 8 times higher than the nozzle diameter of the FDM printer. This is reflected in the quality of the final results: SLA-printed models incorporate fewer microbubbles and show a smoother inner wall We added this description in Section 3.1 and highlighted the difference in the related figure (former 3.2).

  1. In addition, polishing was added as a post-care procedure for FDM-manufactured molds. How will this affect the surface quality and overall accuracy of the printed item in comparison to the CAD design?

We minimized manual intervention as much as possible but polishing is necessary to reduce the traces left by the supports on the surface (for SLA and FDM models) and to smooth the surface for FDM models. The operation was performed with fine and extra-fine sandpaper to avoid compromising the quality of the model. Moreover, we liquefied the resin and filaments left over from the removal of the supports and used the resulting liquid to counterbalance the polishing step. We clarified this aspect in Section 2.3.

  1. And what is the smallest feature size that can be produced using either printing technique in terms of the inner diameter of the finished tubular structure?
  2. We minimized manual intervention as much as possible but polishing is necessary to reduce the traces left by the supports on the surface (for SLA and FDM models) and to smooth the surface for FDM models. The operation was performed with fine and extra-fine sandpaper to avoid compromising the quality of the model. Moreover, we liquefied the resin and filaments left over from the removal of the supports and used the resulting liquid to counterbalance the polishing step. We clarified this aspect in Section 2.3.

  1. For Section 3.3, can you provide some insight into the variation between different sample parts observed in the strain-stress curve? is it because of material inconsistency or manufacturing defects in the dog-bone samples?

Dog bone samples were obtained by die-cutting to ensure uniform thickness. However, since silicone has a low elastic modulus, die-cutting could generate defects on the sides. This could explain the difference in tensile strength at break. In any case, we are interested in the elastic part of the curve, since our models will work at small deformations, and this part of the curve matches very well between the samples. We added a clarification about this in Section 3.3.

  1. Given the silicone was a commercially available material, how does the outcome compare to the material attributes the vendor stated?

The average tensile strength and elongation at break are indeed close to those indicated by the vendor and reported in Table 3. From the mechanical tests on the silicone samples, we observed a nonlinear elastic behaviour with Young's modulus ranging from 0.5 to 2 MP, which could not be inferred from the vendor information about this material. We added a clarification about this in Section 2.4, 3.3, 4

  1. Furthermore, is the material property as determined from a bulk material a reliable indicator of the tubular thin-wall structure?

The thickness of the dog-bone samples is close to that of the final models, so this cannot be considered a test on the bulk material. We added a clarification about this in Section 2.4.

  1. For Section 3.4, based on the marker in the photo, it is unclear whether measurements were taken from a cross-section that was vertical to the substrate plane or perpendicular to the local axis.

The reviewer is correct, there was a mismatch between the figures and the description in the text. The measurements are taken from a cross-section vertical to the substrate. We corrected it in Section 3.4.

  1. For figure 11, it would be preferable to include the RI results for each figure as a comparison.

The image was fixed according to the reviewer’s comment and substituted.

  1. Additionally, a fluid test that compares the suggested vascular phantoms to in vivo models can be used to show how well they work fluidically. I'm asking because a tubular structure's fluid mechanics may also be impacted by the surface hydrophilic characteristic.

We plan to use our models for in vitro studies as a future development of the work presented in this article. Therefore, we are currently gathering the instrumentation to perform studies involving the use of flowing fluids within our models. As we plan to deploy an endovascular device within the models (flow diverter or coils used for the treatment of cerebral aneurysms), the friction coefficient of the model will also play an important role in the results, as we indicate in the Conclusions. We thank the reviewer for this suggestion and will take it into account for our future developments.